# Sterol Regulation of Development and 20-Hydroxyecdysone Biosynthetic and Signaling Genes in *Drosophila melanogaster*

**DOI:** 10.3390/cells12131739

**Published:** 2023-06-28

**Authors:** Di Wen, Zhi Chen, Jiamin Wen, Qiangqiang Jia

**Affiliations:** 1College of Biological Science and Agriculture, Qiannan Normal University for Nationalities, Duyun 558000, China; wendyy2007@163.com; 2Guangdong Provincial Key Laboratory of Insect Development Biology and Applied Technology, Institute of Insect Science and Technology, School of Life Sciences, South China Normal University, Guangzhou 510631, China; 2021023040@m.scnu.edu.cn; 3Guangmeiyuan R&D Center, Guangdong Provincial Key Laboratory of Insect Developmental Biology and Applied Technology, South China Normal University, Meizhou 514779, China

**Keywords:** dietary sterol, biosynthesis, 20-hydroxyecdysone, *Drosophila melanogaster*

## Abstract

Ecdysteroids are crucial in regulating the growth and development of insects. In the fruit fly *Drosophila melanogaster*, both C_27_ and C_28_ ecdysteroids have been identified. While the biosynthetic pathway of the C_27_ ecdysteroid 20-hydroxyecdysone (20E) from cholesterol is relatively well understood, the biosynthetic pathway of C_28_ ecdysteroids from C_28_ or C_29_ dietary sterols remains unknown. In this study, we found that different dietary sterols (including the C_27_ sterols cholesterol and 7-dehydrocholesterol, the C_28_ sterols brassicasterol, campesterol, and ergosterol, and the C_29_ sterols β-sitosterol, α-spinasterol, and stigmasterol) differentially affected the expression of 20E biosynthetic genes to varying degrees, but similarly activated 20E primary response gene expression in *D. melanogaster* Kc cells. We also found that a single dietary sterol was sufficient to support *D. melanogaster* growth and development. Furthermore, the expression levels of some 20E biosynthetic genes were significantly altered, whereas the expression of 20E signaling primary response genes remained unaffected when flies were reared on lipid-depleted diets supplemented with single sterol types. Overall, our study provided preliminary clues to suggest that the same enzymatic system responsible for the classical C_27_ ecdysteroid 20E biosynthetic pathway also participated in the conversion of C_28_ and C_29_ dietary sterols into C_28_ ecdysteroids.

## 1. Introduction

Ecdysteroids are assumed to be the principal steroid hormones regulating insect growth and development, especially molting and metamorphosis, including egg hatching, larval–larval molting, and larval–pupal–adult metamorphosis [1,2,3]. Sterols are precursors for steroid hormones. While vertebrates can produce steroid hormones from cholesterol that is synthesized endogenously de novo, insects are sterol auxotrophs, unable to biosynthesize cholesterol, and therefore must rely on dietary sterols for ecdysteroid biosynthesis [4]. Zoophagous insects can obtain cholesterol synthesized by other animals. But, for fungivorous insects, the dietary sterol is mainly ergosterol. And, for herbivorous insects, these dietary sterols are a variety of different phytosterols, such as campesterol, brassicasterol, β-sitosterol, stigmasterol, and α-spinasterol. These phytosterols typically differ from cholesterol in the number and position of double bonds in the tetracyclic sterol nucleus or in the side chain, or by having additional substituents, such as methyl or ethyl groups at C-24 in the side chain (Figure 1) [5].

It has been a long-held view that 20-hydroxyecdysone (20E) is the major and most active form of molting steroid hormone in insects, and a set of genes mediating 20E biosynthesis, includes four cytochrome P450 (CYP) enzymes that encoded by genes in the Halloween family [6]. Cholesterol is a C_27_ sterol which serves as the starting substrate in 20E biosynthesis. In the specialized ecdysteroidogenic organ that called the prothoracic glands, cholesterol is firstly converted into 7-dehydrocholesterol (7-dhC) by Neverland (Nvd) oxygenase (Figure 1A) [7]. Then, 7-dhC is converted into 2, 22, and 25-trideoxyecdysone (ketodiol) by a series of uncharacterized oxidative reactions known as the “Black Box”. 7-dhC is the precursor of uncharacterized intermediates in the Black Box [8]. Shroud (Sro), CYP307A1 (Spook, Spo), and CYP307A2 (Spookier, Spok) act as stage-specific components of the “Black Box”, but their substrates are still unknown so far [6,8,9,10,11]. The 25-hydroxylase CYP306A1 (Phantom, Phm), the 22-hydroxylase CYP302A1 (Disembodied, Dib), and the 2-hydroxylase CYP315A1 (Shadow, Sad) catalyze the three sequential hydroxylations yielding ecdysone (E) from ketodiol [12,13,14,15,16]. The synthesized E, the precursor of 20E, is then secreted into the hemolymph and converted to 20E by the 20-hydroxylase CYP314A1 (Shade, Shd) in the peripheral tissues, such as fat body and midgut (Figure 1A) [10,15]. Then, 20E binds to the ecdysone receptor (EcR) and ultraspiracle (Usp) to form the 20E-EcR-USP complex, leading to the initiation of the 20E-triggered transcriptional cascade. The 20E-EcR-USP complex first induces the expression of a small set of primary response genes that encode transcription factors such as *Broad-Complex* (*Br)*, *Ecdysone-induced protein 74* (*E74)*, *Ecdysone-induced protein 75* (*E75)*, and *Ecdysone-induced protein 93* (*E93)*, which are responsible for the upregulation of a big set of downstream secondary response genes for successful molting or metamorphosis [17,18,19,20,21,22]. These transcription factor genes were termed the “primary response genes” because they are activated and transcribed within minutes after 20E stimulation, without the need for de novo protein synthesis [23].

Although zoophagous insects can obtain cholesterol directly from their diet, many phytophagous insects must dealkyate phytosterols to obtain cholesterol, as phytosterols are largely C_28_ and C_29_ compounds alkylated at the C-24 position (Figure 1B) [3,24,25,26]. In the meantime, some phytophagous insects also directly convert phytosterols into C_28_ or C_29_ ecdysteroids, such as makisterone A (MaA, C_28_) or makisterone C (MaC, C_29_) [3,24]. Previous studies have concluded that C_27_ ecdysteroid 20E is the major steroid hormone in the fruit fly *Drosophila melanogaster* [1]. However, according to a recent study, when *D. melanogaster* were fed with only a single kind of C_28_ or C_29_ sterol with a methyl or ethyl group at C-24, animals exclusively produce C_28_ ecdysteroid [3]. This study showed that the dealkylation of C_28_ sterols is not necessary for ecdysteroid biosynthesis in *Drosophila* [3]. So, insect ecdysteroids are not only C_27_ ecdysteroids (E and 20E) but also C_28_ ecdysteroids, such as Makisterone A (MaA), 24-epi-Makisterone A (24-epi-MaA), 24, 28-dehydromakisterone A (dhMaA), and so on (Figure 1C) [3].

The 20E biosynthesis and signaling pathway has been particularly well investigated in *D. melanogaster* [27]. Although it has been reported that only C_28_ ecdysteroids were produced when fed with different C_28_ or C_29_ dietary sterols in *D. melanogaster*, it remains unknown as to whether there are functional differences between different dietary sterols in *D. melanogaster*, whether the different dietary sterols synthesized ecdysteroids through classical 20E biosynthesis, and how the classical 20E signaling pathway is regulated by C_28_ ecdysteroid. Here, our study revealed that single dietary sterol was enough to support *D. melanogaster* growth and development. Further studies provided preliminary clues to suggest that different dietary sterols were converted into C_28_ ecdysteroids through classical 20E biosynthetic genes, although their expression levels showed some difference. Additionally, we found that C_28_ ecdysteroids might have similar inductive effects on 20E primary response genes. 

## 2. Materials and Methods

### 2.1. Chemicals

Ecdysone (11711), 20-hydroxyecdysone (16145), makisterone A (11739), and α-spinasterol (14197) were purchased from Cayman Chemical, and cholesterol (C8667-5G), 7-dhC (30800-5G-F), ergosterol (E6510-5G), campesterol (C5157-10MG), brassicasterol (B4936-5MG), β-sitosterol (S1270-10MG), and stigmasterol (S2424-1G) were purchased from Sigma-Aldrich (St. Louis, MO, USA). 22-NBD cholesterol (HCQ000646) was purchased from BioFount Beijing Bio-Tech Co. Ltd., Beijing, China. Chloroform (10006818) was purchased from Sinopharm Chemical Reagent Co., Ltd., Shanghai, China. 2-Hydroxypropyl-β-cyclodextrin (HβC) (ST2114-5g) was purchased from Shanghai Beyotime Biotech. Inc. (Shanghai, China) DMSO (60313ES60) was purchased from Yeasen Biotechnology (Shanghai) Co., Ltd. (Shanghai, China).

### 2.2. Cell Culture

*D. melanogaster* Kc cells were maintained in complete medium (Schneider’s insect medium (Sigma-Aldrich, St. Louis, MO, USA) containing 5% fetal bovine serum (HyClone, Logan, UT, USA)) [28,29]. The cell lines were generously provided by Prof. Sheng Li of the South China Normal University, and were originally purchased from the Drosophila Genomics Resource Center (DGRC) (DGRC Stock 1; https://dgrc.bio.indiana.edu//stock/1 (accessed on 6 June 2022); RRID:CVCL_Z834). The day before sterol treatment, Kc cells were washed with PBS, inoculated into 12-well plates, and cultured with Schneider’s complete medium without fetal bovine serum. Each well contained one milliliter of culture medium. Sterols (including 22-NBD cholesterol) were stored in 10 mM concentrations in chloroform solution or in 45% HβC solution, 2 µL was added to culture medium, and the control was added in the same volume as that of chloroform. Ecdysone, 20-hydroxyecdyosne, and makisterone A were dissolved with DMSO to 10 mM. The treatment concentration of all of the chemicals was 200 µM. After 48 h treatment, cells were collected for total RNA extraction. The 22-NBD cholesterol-treated live Kc cells were nuclei-labeled with DAPI (40728ES03, Yeasen Biotechnology) and then directly observed using an Olympus Fluoview FV3000 confocal microscope at 100× magnifications.

### 2.3. Fly Stocks

Wild-type *w*^1118^ flies were available from the Bloomington Stock Center. The fly strain was kept on standard cornmeal/molasses/agar medium at 25 °C under a 12 h/12 h light/dark cycle with 50 ± 5% relative humidity.

### 2.4. D. melanogaster Diets

All experiments were performed at 25 °C. For experiments, flies were reared on lipid-depleted medium (LDM) (10% chloroform-extracted yeast autolysate (73145-500g-F, Sigma-Aldrich, USA), 10% glucose, 1% chloroform-extracted agarose, and 0.015% methyl paraben) which was made as previously described [30]. Sterols were dissolved into chloroform at 10 mM storage concentrations and were added to a final concentration of 200 µM in LDM; the control was added in the same volume as chloroform. Each bottle contained 5 mL of LDM or LDM with sterols.

### 2.5. Developmental Timing Analysis

Eggs were collected at 25 °C for 1 h on the grape-juice agar plates. After rinsing with distilled water for 5 min, 20 eggs were transferred into each bottle containing 5 mL of LDM or LDM with different sterols added (the concentration was 200 µM). Then, after 9 days, pupariation was recorded by counting pupa numbers every 24 h [28].

### 2.6. Total RNA Extraction and Quantitative Real-Time PCR

Kc cells after 48 h of treatment of sterols, and animals 10 days after egg laying or in the white prepupal stage upon treatment of sterols, were collected. Total RNA was extracted using TRIzol (Invitrogen, Waltham, MA, USA) according to the manufacturer’s instructions. Then, total RNA (2 µg) was reversely transcribed into cDNA using Reverse Transcriptase Goldenstar^TM^ RT6 (Tsingke, China). Real-time quantitative PCR (qRT-PCR) was performed using T5 Fast SYBR^®^qPCRMix (Tsingke, Beijing, China) and the Applied Biosystem^TM^ QuantStudio^TM^ 6 Flex Real-Time PCR System. qRT-PCR was performed as described previously, using *rp49* as an internal control [21,22,31]. The primers used for qRT-PCR are listed in Table 1. Three biological replicates were performed.

### 2.7. Immunocytochemistry of Drosophila Tissues

The brain-ring gland complex and fat body of different sterol treatments were dissected in the white prepupal stage in the PBS (pH 7.0). Dissected tissues were fixed and stained with antibodies according to standard procedures. The primary antibodies used included rabbit anti-Nvd (1:200), rabbit anti-Sad (1:200, SHW-101AP, FabGennix, Frisco, TX, USA), and rabbit anti-Shd (1:200, SHD-101AP, FabGennix). The secondary antibody was Alexa Flour^TM^ 488 goat anti-rabbit IgG (1:200, A-11008, ThermoFisher, Waltham, MA, USA). The rabbit polyclonal antibody against *Drosophila* Nvd was generated via the immunization of rabbit with epitope YNHKRDTINRLRKLK by ChinaPeptides (Shanghai, China). DAPI was used for nuclei labeling. Fluorescence signals were detected using an Olympus Fluoview FV3000 confocal microscope at 40× magnifications.

### 2.8. Measurement of Free Cholesterol Level

Briefly, twenty white prepupal-stage animals were collected and total cholesterol was extracted with 200 μL of chloroform: isopropanol: IGEPAL CA-630 (7:11:0.1) using a microhomogenizer. Then, total cholesterol was measured via GC-MS according to a previously published GC-MS method for the quantification of free cholesterol [32].

### 2.9. Statistical Analysis of Data

All of the data were presented as the mean ± standard deviation of three independent experiments, and were analyzed using Student’s *t*-test and ANOVA. Asterisks indicate a significant difference as calculated using the two-tailed unpaired Student *t*-test (*, *p* < 0.05, **, *p* < 0.01). For ANOVA, the bars labeled with different lowercase letters are significantly different (*p* < 0.05).

## 3. Results

### 3.1. Dietary Sterols Inhibit the Most 20E Biosynthetic Genes but Activate the 20E Signaling Pathway in D. melanogaster Kc Cells

The *D. melanogaster* Kc cell line is derived from embryos and expresses many genes involved in the 20E biosynthesis and signaling pathway (http://flybase.org/ (accessed on 22 August 2020)). The cytochorome P450-18A1 (CYP18A1) inactivated 20E by catalyzing 26-hydroxylation of 20E. We further tested whether Kc cells could produce 20E by overexpressing *Cyp18a1* and *Shd*. The overexpression of *Cyp18a1* in the Kc cells reduced the expression levels of two 20E primary response genes *E74* and *E75*; instead, the overexpression of *Shd* increased the expression of the above two genes (Appendix A). This indicates that Kc cells can produce 20E, perhaps in very small amounts. Dietary sterols, serving as precursors of ecdysteroids, may regulate ecdysteroidogenic genes and 20E primary response genes. First, two types of sterol solvents, chloroform and HβC, which can act as delivery vehicles, were compared for their efficiency in carrying sterols into cells. Using 22-NBD cholesterol as a fluorescent probe, the results showed that HβC worked better and more sterols in the HβC group were localized in the plasma membrane (Figure 2). Then, we investigated the effect of different single dietary sterols and three ecdysteroids (E, 20E, and MaA) on eight 20E biosynthetic genes (*Nvd*, *Sro*, *Spo*, *Spok*, *Phm*, *Dib*, *Sad*, and *Shd*) and four primary response genes (*Br*, *E74*, *E75*, and *E93*) in Kc cells via qRT-PCR; the control group was treated with chloroform, as seen in Figure 3A, 45% HβC, as seen in Figure 3B, and DMSO, as seen in Figure 3C. As expected, all three ecdysteroids promoted the expression of 20E primary response genes (>10 folds), and the effect of E was weaker than 20E and MaA (Figure 3C). Except for *Nvd* and *Phm*, the expression of the other 20E biosynthetic genes was activated by three ecdysteroids (>2.0 folds). Upon all eight forms of sterol treatment, the expressions of *Nvd* and *Spok* were decreased. The expression patterns of *Sro*, *Spo*, *Dib*, and *Sad* were similar: they were highly expressed (>1.5 folds) upon brassicasterol or α-spinasterol treatment, but were less expressed (<0.8 folds) upon other forms of sterol treatment. The expression of *Phm* was a little different from the above four genes: it was down-regulated in brassicasterol and stigmasterol, while it was induced by other forms of sterol treatment. It seemed that *Shd* expression was not significantly affected by these eight dietary sterols (Figure 3A,B). 

For these four primary response genes of 20E signaling, different dietary sterols induced 20E response gene expression, and the induction effect of brassicasterol was the most pronounced, while the induction effects of cholesterol and campesterol were relatively weak. These cell treatment experiments preliminarily indicated that 20E biosynthetic genes in *D. melanogaster* were regulated by dietary sterols and different dietary sterols could be utilized by *D. melanogaster* to biosynthesize ecdysteroids. It is worth noting that there are slight differences between chloroform-dissolved sterols and HβC-dissolved sterols, probably because of the different amounts of sterols absorbed by the cells. As shown in Figure 2, the cells took up more sterols when HβC was used as a solvent.

### 3.2. Single Dietary Sterol Is Enough to Support D. melanogaster Growth and Development

To answer whether different dietary sterols can support the growth and development of *D. melanogaster*, we observed their larval development on LDM or LDM supplemented with single dietary sterols. Most flies fed with LDM supplemented with dietary sterols completed their development from egg to adult. However, the development of those fed with LDM was arrested in the first instar larval stage (Figure 4A). Although an absence of dietary sterol arrested larval development, the larvae would live for more than six days without further development and continued to feed until they died. These results suggest that *D. melanogaster* might be capable of synthesizing ecdysteroids from many kinds of dietary sterols.

To better understand the effects of different dietary sterols on the growth and development of *D. melanogaster*, we compared the developmental timing and percentage of pupariation. As shown in Figure 3B, compared with the pupariation time of cholesterol-fed flies, ergosterol and α-spinasterol resulted in pupariation ∼12 h earlier and ∼12 h later, respectively, with a significantly difference (Figure 4B).

Recently, Lavrynenko et al. revealed that *D. melanogaster* only synthesized C_28_ ecdysteroids when they were reared on LDM with C_28_ or C_29_ dietary sterols [3]. This indicated that *D. melanogaster* cannot dealkylate C_28_ sterols to produce cholesterol. Our free cholesterol measurement experiments further confirmed that cholesterol was not produced when *D. melanogaster* were fed with C_28_ or C_29_ dietary sterols (less than 5 ng/animal) (Figure 4C). The content of free cholesterol is about 750 ng/animal in the cholesterol-fed group and is about 15 ng/animal in the 7-dhC-fed group. Compared to other dietary sterols, higher concentrations of free cholesterol can be detected in the 7-dhC-fed group, maybe coming from an impurity in the commercial 7-dhC. Taken together, these results suggest that single dietary sterol was sufficient to support *D. melanogaster* growth and development.

### 3.3. Effect of Dietary Sterols on 20E Biosynthetic Genes in D. melanogaster

Subsequently, we explored the effects of different dietary sterols on 20E biosynthetic genes in *D. melanogaster* reared on LDM spiked with different sterols in the white prepupal stage via qRT-PCR (Figure 5). The results showed that the expression levels of *Phm* and *Dib* were relatively constant in flies reared on diets with different dietary sterols. The expression levels of *Nvd* were significantly increased in campesterol- and α-spinasterol-reared flies, while they were significantly reduced in 7-dhC- and ergosterol-reared flies. The expression levels of *Spok* were significantly increased in β-sitosterol- and α-spinasterol-reared flies. The expression levels of *Sad* were significantly increased in campesterol-, α-spinasterol-, and stigmasterol-reared flies, while they were significantly decreased in brassicasterol-reared flies. The expression levels of *Shd* were significantly increased in α-spinasterol-reared flies. Interestingly, there was an overall down-regulation in biosynthetic gene expression above 20E in ergosterol-reared flies.

Considering the significant changes in *Nvd* and *Sad* upon different sterol treatments, we examined their protein levels in the prothoracic gland. The results showed that Nvd and Sad protein abundance became greater in campesterol-, α-spinasterol-, and stigmasterol-reared flies, and Nvd protein abundance became less in 7-dhC-reared files, which is consistent with the qRT-PCR results (Figure 6, left and middle columns, Appendix A). The Shd protein levels in the fat body became more abundant in α-spinasterol-reared flies and less abundant in ergosterol-reared flies, which is also consistent with the qRT-PCR results (Figure 6, right column, Appendix A).

These results suggest that, except for *Phm* and *Dib*, enzymes participating in the 20E biosynthesis in flies had different expression patterns after different sterol treatments, indicating that the catalytic ability of these enzymes to different sterol treatments was different. The expression of most 20E biosynthetic enzyme genes was decreased after ergosterol treatment but increased after α-spinasterol treatment, indicating that ergosterol might be a more suitable substrate for ecdysteroid biosynthesis compared with other dietary sterols.

### 3.4. Regulation of Dietary Sterols on 20E Signaling Pathway in D. melanogaster

Our previous studies confirmed that dietary sterols induce the 20E signaling pathway in *D. melanogaster* Kc cells. To further investigate whether *D. melanogaster* reared on single dietary sterols need relatively highly expressed 20E signaling primary response genes to finish larval–larval molting and larval–pupal metamorphosis, we analyzed the relative expression of four 20E signaling primary response genes in first instar larvae (about 48 h after egg laying) and those in the white prepupal stage.

In the first instar larvae, compared with the chloroform-treated group of *D. melanogaster*, the expression levels of *Br*, *E74*, *E75*, and *E93* in the sterol-treated groups were significantly higher. Moreover, compared with other sterol-treated groups, the ergosterol-treated group showed higher expression and the α-spinasterol-treated group showed lower expression. In the white prepupal stage, compared with the cholesterol-reared group, *Br*, *E74*, *E75*, and *E93* expression in the other dietary sterol-treated groups did not show a significant difference (Figure 7B). These results imply that the C_28_ ecdysteroids synthesized from C28 and C29 sterols have similar functions with 20E, which is consistent with the above Kc cell results, showing that there were no obvious differences between 20E and MaA upon the induction of 20E primary response genes.

## 4. Discussion

In the present study, we have determined that dietary sterols are essential for *D. melanogaster* growth and development, and a single kind of dietary sterols can be utilized by flies to complete their growth and development. This is consistent with previous studies that have shown that insects are unable to synthesize sterols and must obtain dietary sterols from their food to complete their development [33,34]. As we know, cholesterol is not only a steroid hormone precursor but also an important component of cell membranes and is needed for lipid modifications on proteins [25,29]. When fed with LDM only, *D. melanogaster* larvae are arrested in the first instar larval stage. Conversely, those reared on LDM supplemented with one of eight different dietary sterols can complete adult development. Free cholesterol measurement experiments using GC/MS methods showed that almost no cholesterol was produced in the other seven dietary sterol-fed animals (Figure 4C), which demonstrates that these sterols are sufficient to undertake various functions of cholesterol in *D. melanogaster*. Moreover, we found that α-spinasterol-fed larvae had remarkably delayed development, and pupariated 12 h later than the pupariation time point of the cholesterol-fed flies. Thus, we hypothesized that α-spinasterol might not be a proper starting sterol for ecdysteroid biosynthesis. Consistent with this speculation, we found that most 20E biosynthetic genes were upregulated upon α-spinasterol treatment, both in Kc cells and in insects (Figure 3, Figure 5 and Figure 6). This may mean that more enzymes were required for α-spinasterol to convert into ecdysteroids. Our data also showed that, compared to other sterols, ergosterol, a C_28_ fungal sterol, induced a lower expression of 20E biosynthetic genes. We hypothesized that 20E synthetic enzymes may have high catalytic efficiency for converting ergosterol into C_28_ ecdysteroids, or there might be other unknown enzymes involved in the conversion process from ergosterol to C_28_ ecdysteroids.

According to our results, there were significant differences in the induction and reduction patterns of ecdysteroidogenic genes between in vitro and in vivo experiments. For example, the expression of *phm* was not reduced by 7-dhC in vitro but was reduced in vivo. These differences maybe arise due to two reasons. One is that Kc cells grew and proliferated normally under sterol deprivation [29,35], while *D. melanogaster* had to obtain sterols (Figure 4). This led to their different responses to sterol treatments. The other is that Kc cells only produced a very small amount of 20E, while *D. melanogaster* had specialized ecdysteroidogenic organs (the prothoracic glands) to produce relatively higher amounts of 20E. The amounts of sterol ingested by the cells also had an effect on the expression changes in ecdysteroidogenic genes, as shown in Figure 2 and Figure 3A,B. 

It has been reported that *D. melanogaster* can produce four different active ecdysteroids, including 20E, MaA, *epi*-MaA, and dhMaA, using a new LC-MS/MS method [3,36]. Moreover, *Drosophila* only produced C_28_ ecdysteroids when they were reared on phytosterols or fungal sterols [3]. Although the biosynthetic pathway of the C_27_ ecdysteroid 20E from cholesterol is well understood, the process from the C_28_ or C_29_ dietary sterol as a precursor to C_28_ ecdysteroids has remained unknown. Our data showed that all classical 20E biosynthetic genes were expressed when *D. melanogaster* were reared on phytosterols or fungal sterols, implying that C_28_ ecdysteroids may be directly synthesized from C_28_ dietary sterols by the well-known 20E biosynthetic pathway. Considering that *D. melanogaster* fed on C_29_ dietary sterols also only produced C_28_ ecdysteroids, the demethylation must have happened in the 20E biosynthetic pathway, or maybe happened before they entered the classical 20E biosynthetic pathway (pathway ① in Figure 8). Alternatively, C_29_ dietary sterols may have entered into the 20E biosynthetic pathway first, and then the demethylation may have happened at an intermediate step (pathway ② in Figure 8). Above all, based on the above results, it can be preliminary concluded that C_28_ and C_29_ dietary sterols were converted into C_28_ ecdysteroids through the same ecdysteroidogenic gene network of the 20E biosynthetic pathway.

It has been suggested that some insects such as the hemiptera Oncopeltus fasciatus, *Podisus maculiventirs*, and the hymenopteran honey bee, *Apis mellifera*, can utilize the C_28_ ecdysteroid, MaA, as the major molting hormone rather than the more common C_27_ hormone, 20E [37]. A subsequent study examined that the EcR and Usp, which were critical for initiating the primary response gene expression, showed similar affinities for MaA and 20E [26]. MaA was found in *D. melanogaster* a few decades ago [38]; nevertheless, it is still unknown as to how C_28_ ecdysteroids regulate the molting and metamorphosis of *D. melanogaster*. Our research has demonstrated that the C_28_ ecdysteroids from phytosterol and fungal sterols as precursors can also activate the primary response genes of 20E signaling to regulate the growth and development of flies, and MaA showed indistinguishable physiological activity with 20E in inducing the expression of primary response genes *Br*, *E74*, *E75*, and *E93* (Figure 2 and Figure 6). 

In this study, it was found that brassicasterol is capable of supporting *D. melanogaster* larval growth to pupae, with a comparatively slow developmental speed (Figure 3B). But, as Lavrynenko et al. report, only a few individuals (about 5%) can survive to pupae [3]. This discrepancy may come from the different sterol concentrations in two experiments: it was 80 μg/mL in our assay and only 6 μg/mL in their assay. According to Martin’s results, the optimal concentration for cholesterol is up to 1.6 mg/mL [39]. Another study recommended that the cholesterol concentration in the medium should be 100–300 μg/mL [40]. So, the sterol concentration in this assay should be more suitable for *D. melanogaster* growth and development. Another possible reason is that other unknown sterols in commercial brassicasterol were utilized for the *D. melanogaster* to biosynthesize 20E; its purity is 98% according to the manufacturer’s instructions.

In summary, the present study is the first to find that there are differences related to the expression of 20E biosynthetic enzymes after different sterol treatments in vitro and in vivo given to *D. melanogaster*. Moreover, ecdysteroids synthesized from any kind of dietary sterols triggered almost the same 20E signaling activity. These results suggest that *D. melanogaster* possess a strong environment adaptation ability.

## Figures and Tables

**Figure 1 cells-12-01739-f001:**
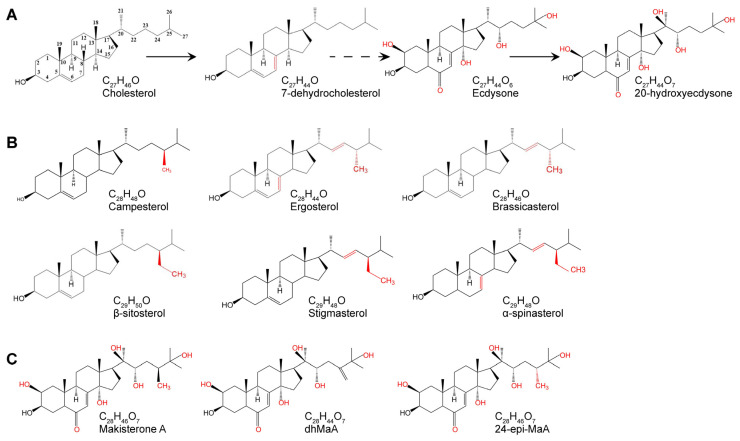
Structures of different sterols and ecdysteroids. (**A**) Structures of C_27_ sterols and ecdysteroids. (**B**) Structures of C_28_ and C_29_ dietary sterols. (**C**) Structures of C_28_ ecdysteroids.

**Figure 2 cells-12-01739-f002:**
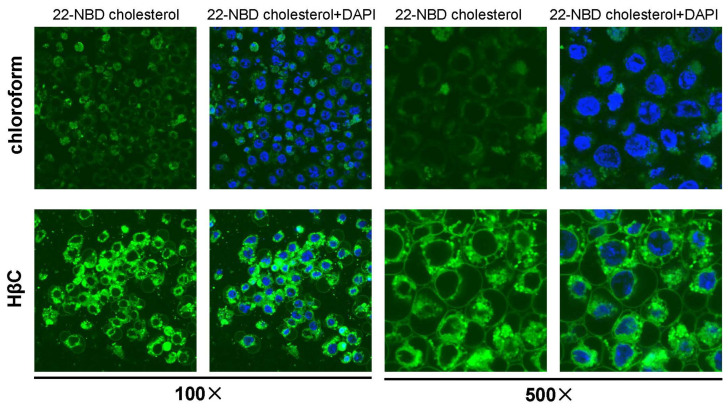
The comparison of the uptake efficiency of Kc cells to chloroform-dissolved 22-NBD cholesterol and HβC-dissolved 22-NBD cholesterol.

**Figure 3 cells-12-01739-f003:**
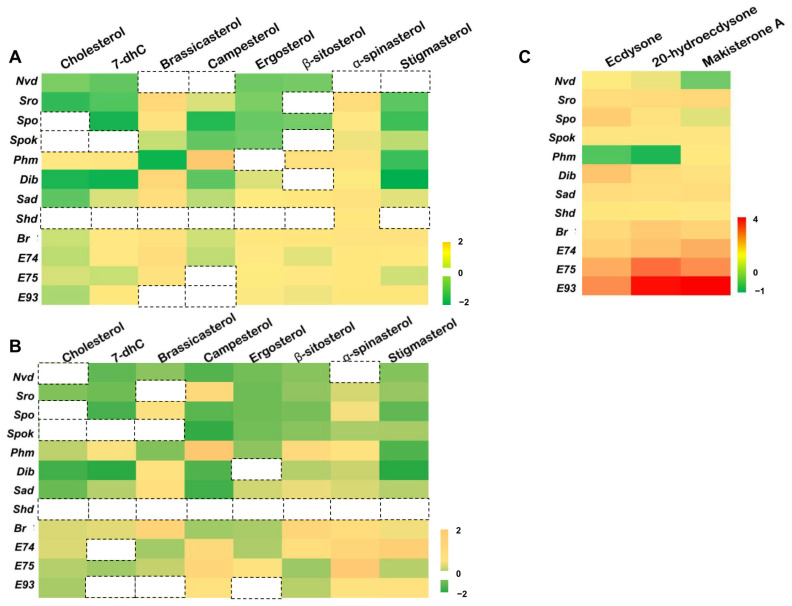
A heatmap of the relative expression of 20E biosynthetic and response genes in Kc cells treated with different sterols. Cholesterol, 7-dehydrocholesterol (7-dhC), brassicasterol, campesterol, ergosterol, β-sitosterol, stigmasterol, and α-spinasterol belong to dietary sterols. E, 20E, and MaA were used to compare the effects of different ecdysteroids. The control group in (**A**) was treated with chloroform, that in (**B**) was treated with 45% HβC, and that in (**C**) was treated with DMSO. Green indicates the lowest expression and red indicates the highest expression. The fold changes in gene expression levels were log2 transformed.

**Figure 4 cells-12-01739-f004:**
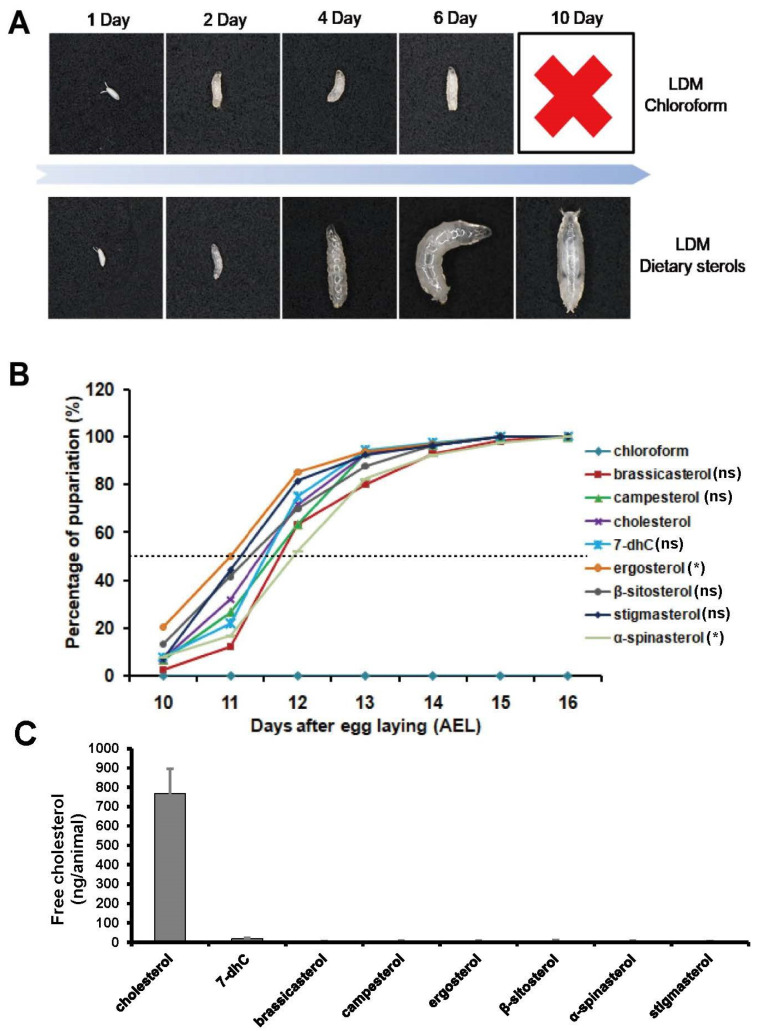
Development of Drosophila reared on diets with controlled composition of sterols. (**A**) Dietary sterol depletion arrests larvae growth and development. LDM with chloroform is the control group. The red cross means death. (**B**) Developmental timing and percentage of pupariation. In each group, 100 animals were measured. Asterisks indicate a significant difference, as calculated using two-tailed unpaired Student’s *t*-test (* *p* < 0.05). “ns” means “no significant difference”. (**C**) Total cholesterol content of whole body of Drosophila reared on LDM supplemented with individual sterols.

**Figure 5 cells-12-01739-f005:**
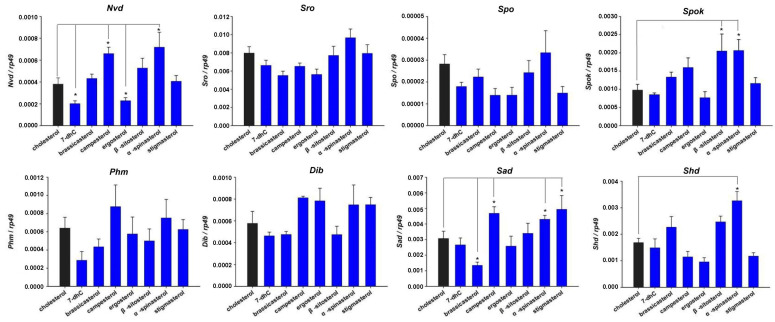
Expression changes in Drosophila 20E biosynthetic genes upon different dietary sterol treatments. Cholesterol was added as a control (black), while other dietary sterols (blue) were used to be compared with it. Data are reported as means ± standard deviation of three independent biological replications. The relative expression was calculated based on the value of the reference genes. Asterisks indicate a significant difference, as calculated using two-tailed unpaired Student’s *t*-test (* *p* < 0.05).

**Figure 6 cells-12-01739-f006:**
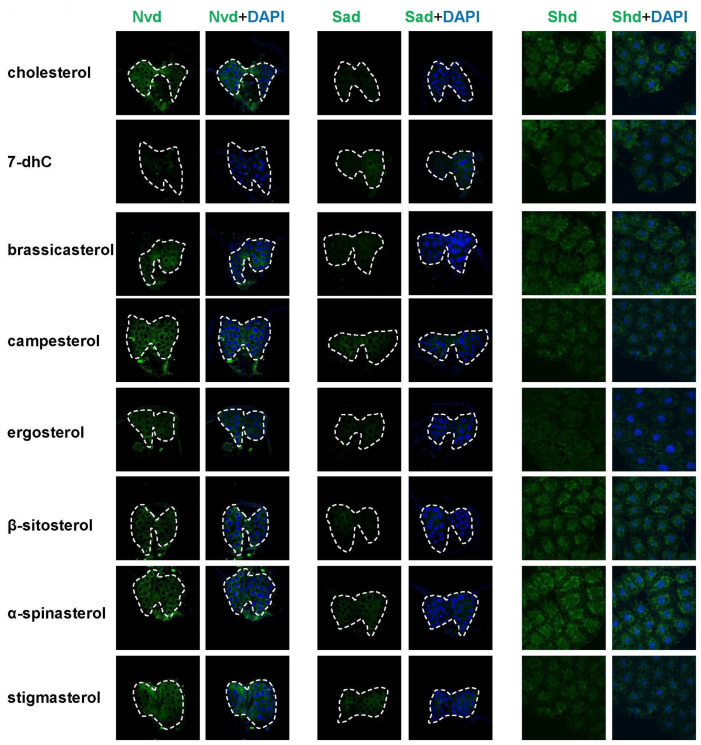
Immunostaining of 20E biosynthetic genes in the prothoracic gland (Nvd and Sad) and fat body (Shd).

**Figure 7 cells-12-01739-f007:**
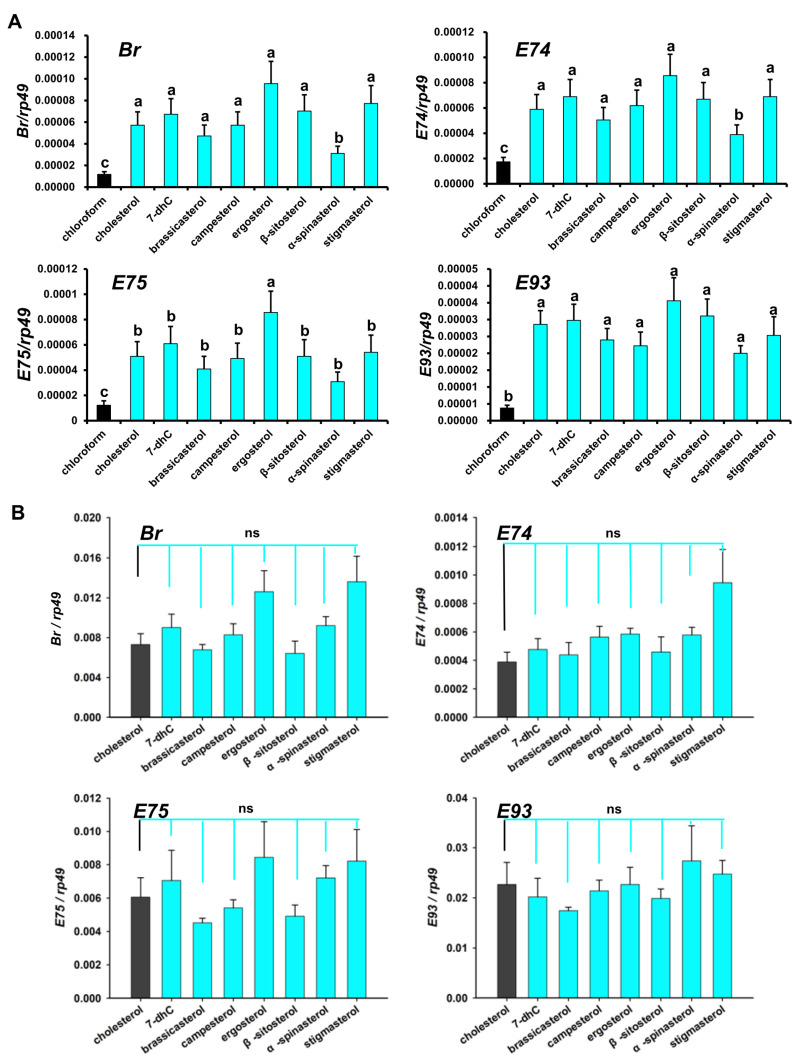
Expression changes in Drosophila 20E signaling pathway genes upon different dietary sterol treatments. (**A**) The gene expression changes in the first instar larvae, about 48 h after egg laying. The chloroform-fed group was the control (black). The bars labeled with different lowercase letters are significantly different (*p* < 0.05) (**B**) The gene expression changes in the white prepupal stage. Cholesterol was added as a control (black), while other dietary sterols (green) were used to be compared with it. Data are reported as means ± standard deviation of three independent biological replications. The relative expression was calculated based on the value of the reference genes. “ns” means “no significant difference”.

**Figure 8 cells-12-01739-f008:**
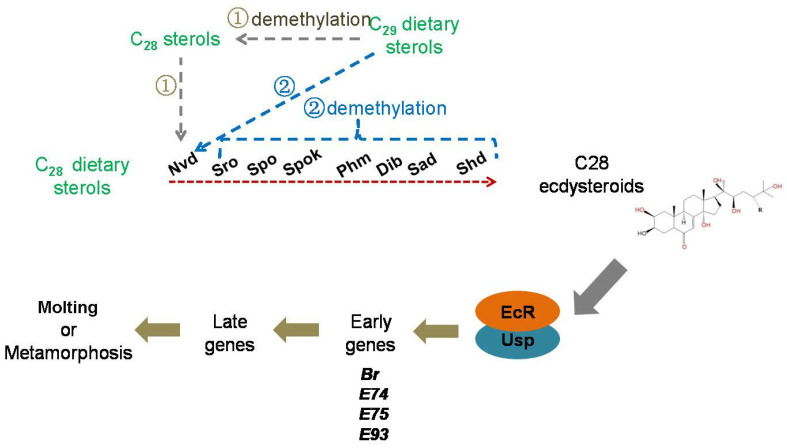
Schematic description of hypothesis for C28 ecdysteroid biosynthesis and signaling pathway in Drosophila. Proposed pathway for biosynthesis of C28 ecdysteroid, catalyzed by Halloween genes.

**Table 1 cells-12-01739-t001:** Primer used for qRT-PCR in this study.

Gene Name	Forward (5′-3′)	Reverse (5′-3′)
*Nvd*	GGGGAGATTGACGATACATT	TTTGAGCCCAACTGCCATTA
*Sro*	CCACAACATCAAGTCGGAAGGAGC	ACCAGGCGAATGGAATCGGG
*Spo*	GCTGCGATATTCATCCTCCC	GCTTTGTTCCTTTGACGGTTC
*Spok*	ATCAACTATTGGACCAGCTTAC	TAATTGAACCGAACTGAACAC
*Phm*	GGATTTCTTTCGGCGCGATGTG	TGCCTCAGTATCGAAAAGCCGT
*Dib*	TGCCCTCAATCCCTATCTGGTC	ACAGGGTCTTCACACCCATCTC
*Sad*	CCGCATTCAGCAGTCAGTGG	ACCTGCCGTGTACAAGGAGAG
*Shd*	TACCCATTCCGACCTCCCACT	TGGTCCTTGCTCGTTCACAATT
*BrC*	CTCAACACGCACACCCAAT	GCTGAAGAGGGTCGAGGAG
*E74*	GCCGGACATGAACTACGAGA	CTTGGGCACATCCACGAAC
*E75*	CCTCAAGCAGCGCGAGTT	GCGATTTCCTTGTGGGTCT
*E93*	GCTGAAGAATGTATGGGTCG	GGGATTGCTCTGGCTGAT
*Rp49*	GACAGTATCTGATGCCCAACA	CTTCTTGGAGGAGACGCCGT

## Data Availability

The data presented in this study are available on request from the corresponding author. The data are not publicly available due to further analysis will be conducted.

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
