# Peer review of "Sterol Regulation of Development and 20-Hydroxyecdysone Biosynthetic and Signaling Genes in Drosophila melanogaster"

_cells, 2023, doi:10.3390/cells12131739_

Round 1

Reviewer 1 Report

Ecdysteroids are very important for insect development and 20-hydroxyecdysone (20E) is the major C27 steroid hormone. 20E biosynthesis has been well studied in Drosophila. Cholesterol is a primary C27 sterol and is converted to ecdysone (E) via many enzymatic reactions in the prothoracic gland. Secreted ecdysone (C27) is converted into 20E in peripheral tissues. While some insects can obtain cholesterol directly from their diet, others insects cannot because their major dietary sterols (phytosterols) are C28 and C29 sterols. It is known that some phytophagous insects produce C28 ecdysteroids from phytosterols and a single phytosterol is sufficient for its synthesis. However, it is unclear how phytosterols are converted C28 ecdysteroids and whether the classical 20E biosynthesis genes are involved in C28 ecdysteroids synthesis.

In this study, Wen et al. showed the treatment Kc cells with a single different C28 sterol changes the expression levels of 20E biosynthetic and response genes in a steroid-type dependent manner (Fig 2). Interestingly, a single C28 or C29 sterol was sufficient to induce larval growth without cholesterol production in D. melanogaster (Fig 3). Subsequently, authors showed that the feeding of a single different C28 or C29 sterol changes the different 20E biosynthetic genes by qPCR (Fig 4), and immunostaining (Fig 5). Interestingly, among phytosterols, feeding with alpha-spinasterol (C29) delayed development with the upregulation of 20E synthetic genes. On the other hand, ergosterol (C28) resulted in faster pupariation with downregulated 20E synthetic genes. In contrast to the response of 20E biosynthetic genes upon phytosterols, all single C28 or C29 sterols did not change the expression levels of response genes (Fig 6). These results might implicate that C29 sterols are not effective to synthesize C28 ecdysteroid compared to C28 sterols, but these the animal finds a way to generate sufficient ecdysteroid to undergo molting, regardless of the source. In conclusion, the authors demonstrated how a single phytosterol is sufficient got molting and pupariations, and found some interesting gene expression changes that accompanies this phenomenon.

I believe the data presented in this manuscript will be of interest to biologists who works in this specific field, considering that C28 ecdysteroid synthesis pathway is not well studied. However, the paper is a bit descriptive and does not provide direct mechanistic insights. The authors make some conclusions which are not based on experimental evidence (see major point). If the authors make appropriate textual changes and addresses the other comments, I will be happy to recommend this for publication.

Major comments

1.     The authors tend to make statements that are not backed up by data in several places in the paper. For example in Line 216. the authors say, “…indicating that the catalytic ability of these enzymes to different sterol was different”. However, gene expression changes seen upon application of different lipids may have nothing to do with the catalytic activities of the encoded genes on these lipids. Also in Line 277-278 and Figure 7: The authors makes statements like “C29 dietary sterols are demethylated into C28 sterols firstly, and then enter into 20E biosynthetic pathway” but there are no biochemical data to show this is actually occurring in vivo. These statements need to be accompanied by experimental evidence or should be significantly toned down.

2.     Differences in protein levels shown in Fig5 needs to be quantified and statistics needs to be provided for the authors to draw any conclusions from these experiments.

Minor comment

1.  Can the authors commeng on why feeding of 7-dhc (C27), which is downstream of cholesterolin Fig1A, produces cholesterol (Fig3 C)? Is this a bidirectional reaction?

N/A.

Reviewer 2 Report

This paper by Wen et al describes the regulation of steroid hormone production and signaling pathways in response to C28 and C29 sterols in Drosophila melanogaster.  Drosophila are considered to require dietary cholesterol (C27) for their growth, and it is generally thought that this is the major substrate used to make ecdysone and 20-hydroxyecdysone, and that these are the active signaling forms of essential hormones.  Wen et al show interesting results demonstrating that the flies can survive from egg to pupariation when raised in lipid-depleted media supplemented with any of multiple C28 and C29 sterols, and that cholesterol was not significantly produced, suggesting that these dietary sterols may be used in place of cholesterol.  This is a compelling start to the paper.  The authors assayed changes in the transcript and protein levels of classical ecdysteroid-production enzymes in the presence of different sterols, either in cell culture, prepupal whole animals, or in specific prepupal tissue, and they describe some gene expression changes.  Finally, they show that several known ecdysone target genes can be activated in prepupa or cell culture when given C28 or C29 sterols.  The authors conclude that multiple different sterols, besides cholesterol, are sufficient to support Drosophila development and “that the same enzymatic system responsible for the classical C27 ecdysteroid 20E biosynthetic pathway also converts C28 and C29 dietary sterols into C28 ecdysteroids”.  Generally the study is interesting and informative and the results are mostly clear, although some of the conclusions are a bit overstated and should be modified to better reflect the results.  I provide more specific suggestions below.

Major concerns:

-       Related to results in Figure 2 and 4, which show changes in gene expression in cell culture in response to different sterols, it is not clear why the authors expect any of these genes to change in expression in response to ecdysone precursors or the active hormone.  If more is known about feedback or feedforward regulation with C27 sterols, it would be helpful to explain this.

-       The protein expression changes in figure 5 are hard to see and it is not clear if the image acquisitions were appropriately standardized.  Quantification of the staining levels is needed.

-       On page 7, lines 214-216, the authors state “enzymes participating in the biosynthesis 20E in flies had different expression patterns after different sterol treatments, indicating that the catalytic ability of these enzymes to different sterol was different.”  However, there is no reason to conclude the catalytic ability is changed just because the expression level is changed, and the catalytic activity was not assayed.  This statement and similar ones in other parts of the paper need to be toned down.  The abstract states “that the same enzymatic system responsible for the classical C27 ecdysteroid 20E biosynthetic pathway also converts C28 and C29 dietary sterols into C28 ecdysteroids” but this is not directly demonstrated at all, so needs to be changed.  Alternatively, this could be demonstrated if the feeding of certain sterols was not sufficient to support survival to pupariation in null mutants for individual genes of the ecdysteriod 20E biosynthetic pathway. (Another example of an overstated conclusion: page 10, line 275, where they authors explain that the expression of the pathway genes implies that C28 ecdysterols are produced by the same enzymes as C27 ecdysterols – they don’t know this, and there could be additional enzymes.)

-       Figure 6 is hard to interpret for a few reasons.  For one, cholesterol and other sterols aren’t expected to induce expression of the ecdysone-response genes directly (consistent with figure 2), and so here it is presumed the response genes are activated due to the pre-pupal stage.  But we cannot tell if they have.  It is still interesting to show they can be activated (eg, different sterols are sufficient to support the activation), but with just one time point (and the apparently quite low levels), it is hard to know if that is the case.  The authors may want to show different time points when the signaling would be low as a comparison.  The asterisks (mentioned in the legend) also appear to be missing in this figure.

Minor concerns:

-       The first section of the results could be more clearly explained in terms of defining the “dietary sterols” and the “primary response genes” before the experimental results are described.

-       For figure 2, the results are a bit hard to interpret because it is hard to tell the color for no substantial difference (it seems to be a light green).  It would be helpful to make those rectangles white instead of using the color scale, so they could be more easily distinguished.

-       Traditionally, cholesterol is also thought to be needed for lipid-modifications on proteins, so the authors may want to mention other sterols may be sufficient for serving this function as well.

-       The model in figure 7 suggests that C28 sterols could feed directly to the top of the C27 ecdysterol pathway- but isn’t it possible that (potentially unknown) converting enzymes could feed them into some other downstream point?  Wouldn’t ergosterol, for example, be expected to skip the first step by Nvd?

There are multiple places in the text where the English language usage is awkward or incorrect, or sentence structure is incomplete; these issues should be corrected.

Reviewer 3 Report

This paper investigated impact of different dietary sterols for ecdysteroidogenic genes of Drosophila melanogaster. Since D. melanogaster can utilize dietary sterols in addition to cholesterol, the effort to evaluate the importance of distinct sterols is interesting. However, the paper has several shortcomings of experimental design and over-interpreted. I strongly encourage the authors to address the following point.

1. Need more clear evidence about ecdysteroidogenic activity to test sterols  In the last sentence of abstract, author mentioned as “our study suggested that the same enzymatic system responsible for the classical C27 ecdysteroid 20E biosynthetic pathway also converts C28 and C29 dietary sterols into C28 ecdysteroids.”. Authors demonstrated that test sterols could affect on gene expression of ecdysteroidogenic enzymes, but did not show clear evidence to suggest enzymatic activity. In vivo assay of w1118 line just showed that these sterol could be sufficient for larval development. To claim enzymatic activities on test sterols, additional experiments are required. Conversion of test sterols by enzymes in culture cells, for example, conversion of ergosterols by Neverland in culture cells and following biochemical analysis, will help to propose authors’ opinion. Or, using KD fly of any ecdysteroidogenic enzymes and rescue analysis with ecdysteroids could help to demonstrate the necessity of ecdysteroidogenic enzymes for the conversion of test sterols at least. Without more clear data, the discussion of enzymatic activities to test sterols need modification.  In addition, it would be better to include the possibility of additional enzymes for conversion of C28 or C29 sterols into ecdysteroids  in the discussion. 

2. The design of in vitro experiment

In this paper, authors investigated the change of gene expressions after dietary sterol addition to culture cells. There are doubts about uniform distribution of sterols in the cell medium and sterol uptake of culture cells. Authors solved test sterols in chloroform solution and added 2 μl of the solution to the medium. Sterols in 2 μl of chloroform solution might not spread uniformly in the well. The sterols might not be incorporated efficiently in culture cells because test sterols were added to the medium without any career chemicals, such as hydroxypropyl-beta cyclodextrin. If author already had any data to show uniform incorporation of test sterols, it is better to include at least as supplemental data. Using labeled sterols, such as 22-NBD cholesterol will give more direct evidence for uniform incorporation in the culture cells. Without the prerequisite, it is difficult to evaluate the response of Kc cells exposed with test sterols.

3. The following aspects are unclear in the cell culture experiment. Please add the explanation of experimental design more detail.

1) Final concentration of distinct sterols in the culture medium

2) Final concentration of chloroform solution. If author used 96 well plate, the chloroform concentration might be near 2% because authors mentioned 2 μl of chloroform solution was added. The concentration might be toxic for the cell. Authors need to show chloroform have less negative effect for cell survival.

3) The composition of cell medium containing test sterols. If authors use medium containing FBS, the medium had large amount of cholesterol. Under such condition, the function of test sterols might be misleading.

4. Need explanation about differences of gene expression patterns on ecdysteroidogenic enzymes between in vitro and in vivo experiments.

Induction and reduction patterns of ecdysteroidogenic enzymes were different between in vitro and in vivo experiments. For example, phm expression was not reduced by 7dC in vitro but reduced in vivo. I understand such differences often occur, but need explanation what cause the differences in the discussion part.

5.  Fig. 6

No asterisks were found in the figure 6 though the statistical analysis were mentioned in the figure legends. In addition, t-test is not suitable in this case. Multiple comparison test will be better for it.

Some of the sentences were difficult to understand. i.e. line 84 (Do you mean "Sterols were stored as 10 mM concentration in chloroform solution"?) and line 55 ("This study" means Lavrynenko's paper or the study of this manuscript?). It is better to check entire manuscript and use more precise expression. 

Round 2

Reviewer 3 Report

The manuscript has been revised well. The authors performed additional experiments required and that strengthen their data. Description in introduction and discussion were also improved.  I think this paper will be acceptable.